# A 6&1-FEH Encodes an Enzyme for Fructan Degradation and Interact with Invertase Inhibitor Protein in Maize (*Zea mays* L.)

**DOI:** 10.3390/ijms20153807

**Published:** 2019-08-04

**Authors:** Hongbo Zhao, Steffen Greiner, Klaus Scheffzek, Thomas Rausch, Guoping Wang

**Affiliations:** 1College of Horticulture, South China Agricultural University, Guangzhou 510642, China; 2Centre for Organismal Studies Heidelberg, Department of Plant Molecular Physiology, University of Heidelberg, 69120 Heidelberg, Germany; 3Division Biological Chemistry, Innsbruck Medical University, Biocenter, Innrain 80, A-6020 Innsbruck, Austria

**Keywords:** *Zea mays*, fructan exohydrolase, substrate specificity, invertase inhibitor, regulation

## Abstract

About 15% of higher plants have acquired the ability to convert sucrose into fructans. Fructan degradation is catalyzed by fructan exohydrolases (FEHs), which are structurally related to cell wall invertases (CWI). However, the biological function(s) of FEH enzymes in non-fructan species have remained largely enigmatic. In the present study, one maize CWI-related enzyme named Zm-6&1-FEH1, displaying FEH activity, was explored with respect to its substrate specificities, its expression during plant development, and its possible interaction with CWI inhibitor protein. Following heterologous expression in *Pichia pastoris* and in *N. benthamiana* leaves, recombinant Zm-6&1-FEH1 revealed substrate specificities of levan and inulin, and also displayed partially invertase activity. Expression of Zm-6&1-FEH1 as monitored by qPCR was strongly dependent on plant development and was further modulated by abiotic stress. To explore whether maize FEH can interact with invertase inhibitor protein, Zm-6&1-FEH1 and maize invertase inhibitor Zm-INVINH1 were co-expressed in *N. benthamiana* leaves. Bimolecular fluorescence complementation (BiFC) analysis and in vitro enzyme inhibition assays indicated productive complex formation. In summary, the results provide support to the hypothesis that in non-fructan species FEH enzymes may modulate the regulation of CWIs.

## 1. Introduction

Only 15% of higher plants have acquired the ability to convert sucrose into fructans [1]. Fructan biosynthetic enzymes (FBEs, e.g., sucrose:sucrose 1-fructosyltransferase, fructan:fructan 1-fructosyltransferase, sucrose:fructan 6-fructosyltransferase) are evolutionarily related to acid invertases present in all higher plants, but catalyze, depending on the plant species, the synthesis of different types of fructans, differing in the linkage between fructose units (i.e., 6→2 (levan type), 1→2 (inulin type), or branched) [2]. Fructan degradation is catalyzed by fructan exohydrolases (FEHs), which remove terminal fructose units [3,4,5]. FEH enzymes may either be specific for α-(2,1) fructose linkages (1-FEHs: Inulinases), for α-(2,6) fructose linkages (6-FEHs, levanases), or may hydrolyze both types of fructans (6&1-FEHs). Elucidation of the 3D structures of a FEH from chicory [6], a cell wall invertase (CWI) from *Arabidopsis* [7], and a FBE from *Pachysandra terminalis* [8] has shown that all three proteins consist of an N-terminal five-bladed β-propeller domain, followed by a C-terminal domain formed by two β-sheets. According to the current hypothesis, plant FBEs have evolved from vacuolar invertases (VIs) [9], whereas FEHs were thought to originate from CWIs [10,11,12].

Based on the sequence similarities and the common structural fold of acid invertases and FBEs/FEHs, the possibility of an interaction of FBEs and/or FEHs with invertase inhibitors has been investigated for *Cichorium intybus*, a plant species that synthesizes inulin-type fructans. No evidence was found for inhibitory activity of homologous or heterologous inhibitor proteins against 1-SST, 1-FFT, or 1-FEHs, respectively [13]. Later, the solved first structure of a complex between AtCWI1 and NtCIF, a CWI inhibitor of tobacco [14], has revealed the sequence motifs involved in this protein–protein interaction, with these motifs being absent in FBEs and FEHs from *C. intybus*.

In 2003, van den Ende [15] reported, in a seminal paper, the presence of a CWI-related enzyme with FEH activity (6-FEH, AJ508534) in *Beta vulgaris*, a non-fructan species. This report was followed by the characterization of two CWI-related FEHs from *Arabidopsis thaliana*, another non-fructan species, one showing activity only towards fructans with 6→2 linkage (At-6-FEH), the other hydrolyzing fructans with 6→2 and 1→2 linkage, respectively (At-6&1-FEH) [16]. These observations and the analysis of other sequenced plant genomes have subsequently led to the conclusion that possibly many higher plants encode CWI-related enzymes with FEH activity in their genomes. A study on VIs from rice [17] provided evidence that after recombinant expression of two VI isoforms, OsVIN1 and OsVIN2 in *Pichia pastoris*, both enzymes displayed significant FEH activity, indicating that the distinction between CWIs/VIs and FEHs is not clear cut, with both VI isoforms displaying all sequence motifs associated with classical invertases.

While the biological functions of FEHs in fructan species have been extensively studied in the context of the regulation of fructan metabolism [18,19,20], the role(s) of CWI-related enzymes with FEH activity in non-fructan plants has remained largely enigmatic. In their study on *Arabidopsis* FEHs, de Coninck et al. [16] have speculated on the following functions: a) Hydrolysis of short chain fructans (kestoses) which may be formed by non-fructan plants at high sucrose concentration, presumably due to side activities of VIs; b) degradation of fructans present in the exopolysaccharide coat of pathogenic and/or symbiotic bacteria. Furthermore, as under pathogen challenge CWI is often induced, the combination of CWI and FEH would effectively reduce the ability to synthesize extracellular levan formation by bacterial levan sucrase, which requires millimolar concentrations of sucrose as substrate. In 2013, a study on the defective tobacco invertase Nin88 [12] proposed a model in which this defective invertase would saturate non-specific ionic binding sites in the cell wall, thereby preventing non-productive binding of active CWI to the cell wall and allowing free access of its substrate sucrose and/or invertase inhibitor binding.

To gain deeper insight into the role(s) of CWI-related enzymes with FEH activities in plants devoid of fructan synthesizing enzymes, we have explored the expression and possible functions of one as yet non-characterized CWI-related enzyme in maize: Zm-6&1-FEH1 grouping in the same clade with previously confirmed maize CWIs (i.e., Zm-INCW1 and Zm-INCW2). This maize protein displayed FEH activity, albeit with different substrate specificities with respect to levan, inulin, kestoses, and sucrose. We report on their differential expression during plant development and in response to stress exposure. As to their possible functions, we have probed their interaction with invertase inhibitor. The results give further support to the hypothesis of an indirect role in CWI regulation [12].

## 2. Results

### 2.1. Characterization of Maize Cell Wall Invertase-Related Enzyme Zm-6*&*1-FEH1 with Fructan Exohydrolase Activity

Previous work on maize cell wall invertases had identified four putative CWI isoforms, Zm-INCW1-4 [21]. For ZmINCW2, the loss of which causes the miniature seed phenotype, the physiological role has been firmly established, i.e., catalyzing sucrose hydrolysis in the basal endosperm transfer layer [22,23]. Mining the GenBank database for yet non-characterized CWI-related maize cDNAs, we selected one candidate based on its grouping in a phylogenic tree with confirmed maize CWIs (Figure 1). The predicted protein sequence of Zm-6&1-FEH1 (EU971090) displayed the β-fructosidase motif NDPNG/A and the canonical cysteine-containing catalytic site MWECP (Figure 2). In an unrooted phylogenic tree, Zm-6&1-FEH1 grouped with ZmINCW1 and 2 [23], and OsCIN1, a confirmed CWI from rice [24].

To identify the substrate specificities, Zm-6&1-FEH1 was expressed in *P. pastoris* (Figure 3). It displayed a higher activity towards levan, inulin, and kestoses i.e., being more active against fructans as compared to sucrose (Table 1). Thus, for consistency we have assigned it as an FEH enzyme (with some invertase activity). For comparison, we also expressed Zm-6&1-FEH1 transiently in leaves of *N. benthamiana*. Since observations from stable expression of its XFP-fusions in *A. thaliana* leaves or transient expression in onion epidermal cells (Figure 4) have indicated that Zm-6&1-FEH1 enzyme is cell wall-localized, its activity towards levan, inulin, and sucrose, was determined in salt-eluted (i.e., cell wall-bound) protein fractions (Figure 5). To account for the presence of endogenous invertase and/or FEH activities from *N. benthamiana*, mock transformations with *Agrobacterium tumefaciens* bearing the empty vector were performed to correct for this background. Again, its cell wall fraction revealed substantial enzyme activity with levan and inulin as substrates (Figure 5).

### 2.2. Maize 6*&*1-FEH1 Enzyme Exhibits Differential Expression during Plant Development and in Response to Abiotic Stress

To explore the expression of maize *6*&*1-FEH1* gene, steady state transcript levels of *Zm-6*&*1-FEH1* was monitored by qPCR in different plant developmental stages (Figure 6A), in response to abiotic stress (drought or cold treatment), and after exposure to abscisic acid (ABA) (Figure 7). *Zm-6*&*1-FEH1* was expressed in young plants (4-leaf stage) and in mature plants during flowering stage, including early seed development. Interestingly, expression of *Zm-6*&*1-FEH1* was always higher in the basal (growing) part of the leaf as compared to the distal expanded part. This observation was further supported when the expression profile of both genes was compared along the leaf axis in more detail (Figure 7A). Comparing total levan-hydrolyzing activities in the same tissues confirmed the preponderance of cell wall-associated enzyme activities as compared to soluble activities (Figure 6B).

When young maize plants were exposed to drought, cold treatment, or abscisic acid (ABA) application via the root system, the *Zm-6*&*1-FEH1* gene displayed different response profiles. *Zm-6*&*1-FEH1* expression in the basal leaf parts (growing region) was rather reduced by stress (or ABA) exposure, whereas the reverse was observed in the mature parts (Figure 7A). For drought response in mature plants during the flowering and seed setting stage, expression of *Zm-6*&*1-FEH1* was almost not affected (Figure 7B). In summary, the observed gene expression patterns for Zm-6&1-FEH1 enzyme revealed significant modulation during plant development and in response to abiotic stress exposure.

### 2.3. Maize 6*&*1-FEH1 Enzyme is Able to Form Complex with Invertase Inhibitor Protein

In higher plants (dicots and monocots), invertase inhibitors are known to be involved in post-translational regulation of their target enzymes [25]. Since Zm-6&1-FEH1 protein sequence displays significant homology with CWIs (see above), the possibility of an interaction between maize FEH enzyme and invertase inhibitor had to be considered. An interaction of CWI-related FEH enzymes with these inhibitor proteins could be of functional significance, i.e., via interfering with inhibitor-based invertase regulation and/or allowing post-translational control of FEH enzyme activities. Therefore, it was of interest to probe for complex formation between maize FEH enzyme and maize invertase inhibitor, and to determine whether such binding eventually also affected FEH enzyme activity.

To this end, the impact of maize invertase inhibitor on maize FEH enzyme, i.e., the previously described and functionally characterized Zm-INVINH1 [26] was explored. Unfortunately, the yield of soluble Zm-INVINH1 protein was consistently low. We, therefore, extracted the Zm-INVINH1 transiently expressed in leaves of *N. benthamiana*. When recombinant maize 6&1-FEH1 enzyme (expressed in *P. pastoris*, see above) were co-incubated with Zm-INVINH1, Zm-INVINH1 affected the activities of Zm-6&1-FEH1 enzyme (Table 2).

To explore complex formation between Zm-6&1-FEH1 enzyme and Zm-INVINH1, the assumed interaction was first probed via the split-YFP system to independently confirm the inhibitor-FEH interactions in vivo (Figure 8). When Zm-INVINH1 was fused C-terminally with cYFP, and Zm-6&1-FEH1 enzyme C-terminally with nYFP, significant fluorescence complementation was observed for inhibitor-FEH combination. For control, a cell wall-localized protein from *Arabidopsis* (At1g33811) was fused C-terminally with cYFP, this combination displaying no fluorescence complementation with nYFP-fused Zm-6&1-FEH1 enzyme.

### 2.4. Modeling the Interface of a Complex between Zm-6*&*1-FEH1 and Zm-INVINH1

In a further attempt to unravel the likely mode of protein–protein interaction, a modeling approach was used. Based on the known structures of the complex between AtCWI1 and the tobacco invertase inhibitor NtCIF [14], the interaction of Zm-6&1-FEH1 with Zm-INVINH1 was modeled, using AtCWI1 and NtCIF as templates for a corresponding complex of Zm-6&1-FEH1 with Zm-INVINH1 complex. The resulting model is depicted in Figure 9. As expected, the maize invertase inhibitor is predicted to interact with the active site of Zm-6&1-FEH1 via its sequence motif GTPRGRAD that corresponds to the GDPKFAED motif in NtCIF. In the complex, Arg123 of Zm-INVINH1 corresponds to Lys98 of NtCIF and would, in principle, be able to interact with portions of the fructose molecule or Glu243 of Zm-6&1-FEH1. Conspicuously, the counterparts of Arg144 and Glu101 in NtCIF are alanine in the maize inhibitor Zm-INVINH1, suggesting that salt bridges formed by these residues in the AtCWI1-NtCIF complex are absent in the complex of Zm-6&1-FEH1 with Zm-INVINH1. The model provides support for the view that the polar components constituting the second site of interaction may be dispensable for complex stabilization.

## 3. Discussion

In the present study, three CWI-related cell wall-localized maize enzymes displaying FEH activities against kestose, inulin, and/or levan have been partially characterized. Unexpectedly, all three enzymes were able to interact with invertase inhibitors from the same species. The results corroborate the previously raised hypothesis, that FEH enzymes [12] (or “defective invertases”) may interfere with invertase regulation, while their activities against levan do not exclude a defense function against pathogenic bacteria bearing levan in their extracellular matrix [16]. In fact, as CWI may play an important role during pathogen attack, both functions are not mutually exclusive (see below).

Upon recombinant expression in *P. pastoris* or *N. benthaminana*, the substrate specificity observed for Zm-6&1-FEH1 correlated well between the two expression systems, indicating that structural differences in the glycan chains, as generated in both expression platforms [27], did not have major effects on substrate specificities. Similarly, expression of recombinant 6-FEHs from two non-fructan plants, i.e., *Arabidopsis* and sugar beet, in *Pichia pastoris* had revealed essentially the same substrate specificities as compared to the respective native enzymes [15,16]. In a previous study on the impact of glycan chains on substrate specificity (and total enzyme activity) of FEHs and CWIs, Le Roy et al. [28] observed a pronounced effect of a particular N-glycosylation near the cleft between the N-terminal and C-terminal domains (corresponding to residues 348-350 in Zm-6&1-FEH1). While Zm-6&1-FEH1 shows an N-glycosylation site at this position, this is in contrast to 6-FEHs from sugar beet, Arabidopsis and wheat [28].

When comparing the expression profile of maize *6*&*1-FEH1* in different plant tissues at transcript level with the corresponding total FEH activities (as determined with levan as substrate, Figure 6B) of the insoluble (cell wall) fraction, the best correlation was observed for Zm-6&1-FEH1. As this FEH enzyme has a pI of 9.25, the enzyme activity in the insoluble fraction is likely due to ionic interaction of this isoform with the cell wall matrix. Targeting studies have indicated that ZM-6&1-FEH1 enzymes is localized in the cell wall (Figure 4), the pI value suggests that after secretion, Zm-6&1-FEH1 remains entirely immobilized in the cell wall.

The expression patterns observed for *Zm-6*&*1-FEH1* point to possible functions in both shoot and root (Figure 6). The different expression profiles along the leaf axis of three-week-old plantlets, as well as the different responses to ABA treatment, drought, or cold, further support the notion of distinct functions (Figure 6 & Figure 7). *Zm-6*&*1-FEH1* is also expressed during early seedling development (Figure 6), as is the *Zm-INVINH1* [26]. As during this stage of seed filling the CWIs ZmINCW1 and ZmINCW2 are expressed (albeit in different regions of the growing seed [23], it is tempting to speculate that the Zm-6&1-FEH1 enzyme could impact upon CWI regulation via binding invertase inhibitor and/or by interfering with invertase binding to the cell wall matrix as recently suggested by [12].

Modeling the interface of the complex between Zm-6&1-FEH1 and Zm-INVINH1 based on the crystal structure of the AtCWI1-NtCIF-complex, [14] suggested a similar mode of interaction (Figure 9), with the conserved sequence motifs forming the enzyme inhibitory interface. A respective complex model does not involve polar contributions of the second contact area due to amino acid equipment of the involved area. Interestingly, for the fructan-synthesizing species *Cichorium intybus* (chicory), several combinations of its fructan active enzymes (FAZYs, including 1-FEHs, 1-SST and 1-FFT) with homologous or heterologous invertase inhibitors did not indicate inhibitory effects [13], however, protein-protein complex formation had not been addressed.

What could be the physiological role of invertase inhibitor binding to FEH enzyme? While FEH enzyme could be itself under post-translational regulation, similar to CWIs (Table 2), FEH enzyme could also be part of a mechanism involved in fine tuning CWI activities via competing for inhibitor binding. Thus, following the previously raised hypothesis of a functional role of 6-FEHs in non-fructan plants, i.e., being involved in defense against levan-bearing bacterial pathogens [16], expression of FEHs with 2,6-FEH activitiy at the site of pathogen attack could simultaneously increase CWI activity, thereby lowering the apoplastic concentration of sucrose, the substrate for bacterial levansucrase [29,30]. In this way, levan degradation and prevention of synthesis could go hand in hand.

## 4. Materials and Methods

### 4.1. Plant Material and Cultivation

*Zea mays* L. (SEVERUS, KWS) and *Nicotiana benthamiana* L. were grown in the greenhouse at 25 °C under long day conditions (16 h light period; 300 μmol m^−2^·s^−1^). For monitoring gene expression and FEH activity, maize plants were cultivated on soil either for 3 weeks or until flowering stage, followed by manual pollination. Details of watering regime, drought exposure, cold exposure, and abscisic acid treatment are presented in figure legends of individual experiments. For transient transformation of *Nicotiana benthamiana*, leaves from 8 to 12-week-old plants were transformed by leaf infiltration with *Agrobacterium tumefaciens* strain C58C1 cells harboring appropriate plasmids (see below) via a 1 mL needleless syringe. After 48 h, tissue samples were either used immediately or frozen in liquid nitrogen and stored at −80 °C until use.

### 4.2. Preparation of RNA, cDNA Synthesis and cDNA Cloning

Total RNA was extracted with the GeneMATRIX Universial RNA purification Kit (Roboklon, Berlin, Germany). cDNA synthesis was performed immediately after DNase (AppliChem, Darmstadt, Germany) treatment, using AMV-Reverse Transcriptase (Roboklon, Berlin, Germany). Full-length cDNAs of Zm-6&1-FEH1 (EU971090 [31]) and Zm-INVINH1 (XM_008670754 [26]) were cloned from maize leaf cDNA by PCR (35 cycles: 95 °C/30 sec–55 °C/30 sec–72 °C/1 min/1kb; final extension: 10 min), using Phusion High-Fidelity DNA polymerase with GC buffer (Finnzymes, Schwerte, Germany) and corresponding primers presented in Appendix A. PCR products were fully sequenced (Starseq, Heidelberg, Germany) and cloned into pDONR201 or pDONRzeo vector (Invitrogen, Darmstadt, Germany) to obtain the entry clone for the Gateway system (Invitrogen, Darmstadt, Germany). Phylogenic analyses were performed with clustalX2 [32] and MAFFT [33].

### 4.3. Plasmid Cloning

Recombinant FEH and invertase inhibitor proteins: For transient expression of Zm-6&1-FEH1 and Zm-INVINH1 proteins in *N. benthamiana*, the respective coding regions were cloned into the pB7WG2 vector downstream of the 35S promoter (for primers see Appendix A). For generation of *Pichia pastoris* expression plasmid, the coding region of *Zm-6*&*1-FEH1* was amplified by PCR. Amplified PCR fragment and pPICZαA vector (Invitrogen) was digested with EcoRI and XbaI (New England Biolabs, Frankfurt, Germany). DNA fragment was purified using the NucleoSpin Extract II Kit (Macherey-Nagel, Duren, Germany) according to the manufacturer’s instructions. Purified product was ligated using T4 DNA ligase (New England Biolabs, Frankfurt, Germany), with incubation at 14 °C for 16 h. Ligation product was transformed into *E. coli* competent DH5α cells by electroporation. Subsequently, bacterial cells were plated on low salt LB medium supplemented with zeocin (Invitrogen, Darmstadt, Germany) treatment, using AMV-Reverse Transcriptase (Roboklon, Berlin, Germany). Full-length cDNAs o as a selection marker. Positive colonies were used for vector amplification. Constructs for expression of recombinant Zm-INVINH1 protein in *E. coli* were made by cloning its cDNA without signal peptide into pETG10A vectors.

XFP fusions: For expression of FEH protein C-terminally tagged with YFP, coding region of *Zm-6*&1*-FEH1* was cloned into the pB7YWG2 vector. To obtain a Zm-INVINH1 construct C-terminally tagged with RFP, coding region of Zm-INVINH1was cloned into the pB7RWG2 vector.

Tagging FEH and invertase inhibitor for BiFC analysis: For split-YFP analysis, coding region of *Zm-6*&*1-FEH1* was cloned into the pUBC-nYFP vector [34], whereas coding regions of *Zm-INVINH1* or *At1g33811* were cloned into the pUBC-cYFP vector [34]. All primers used in plasmid cloning are presented in Appendix A.

### 4.4. Gene expression Analysis by qPCR

qPCR analysis was performed with the Rotor-Gene Q system (Qiagen) using SYBR Green (S7563, Invitrogen) to monitor dsDNA synthesis. Thermal cycling conditions were identical for all primer pairs: 95 °C/6 min, followed by 40 cycles of 95 °C/20 sec–58 °C/20 sec–72 °C/20 sec, followed by a melt cycle from 50 to 95 °C. To determine primer efficiency, serial dilutions of the templates were conducted for all primer combinations. Each reaction was performed in triplicate, and the amplification products were examined by agarose gel electrophoresis and melting curve analysis. The expression stability of reference genes (*actin*, *ubiquitin*, *GAPDH*, and *tubulin*) were assessed using GeNorm algorithms, and the relative gene expression level was calculated by normalizing to the geometric mean of the reference genes, according to a previously described method [35]. Primers for reference genes and target genes are presented in Appendix A. For each tissue, three independent cDNA preparations were analyzed with three technical replica each.

### 4.5. Expression of Recombinant FEH Proteins in Pichia Pastoris

pPICZα plasmid (see above) of *Zm-6*&*1-FEH1* (and empty vector as a control) was linearized by *Pme*I, and then transformed into *Pichia pastoris* strain X-33 via electroporation. Further selection and protein purification were done as described by Kusch [36], except that 5% (*v*/*v*) methanol was included in *Pichia* induction medium.

### 4.6. Expression of Recombinant Invertase Inhibitors in E. coli

Plasmid construct for expression of Zm-INVINH1 protein (see above) was transformed into *E.coli* strain Rosetta-gami (Novagen, Madison, WI, USA). Purification of recombinant protein followed the procedure of Eufinger [37].

### 4.7. Plant Transformation

Transient expression in *Nicotiana benthamiana* leaves was performed by *Agrobacterium* leaf infiltration as described in Wolf et al. [38]; transformation with P19 served as control to account for *Agrobacterium* transformation-induced induction of endogenous CWI and FEH activities. Transient expression via particle gun bombardment was performed as previously described in Scott et al. [39]: Inner epidermal peels (2 cm × 2 cm) of *Allium cepa* were bombarded, incubated for 48 h, and visualized by CLSM; GFP fluorescence detection required a 3 h treatment with 20 mM PIPES buffer pH 7.0 prior to CLMS analysis.

### 4.8. Protein Extraction from Plant Material and Determination of FEH Activity

Extraction of soluble and cell wall-bound FEH proteins from maize tissues and *Nicotiana benthamiana* leaves essentially followed the protocol described in Link et al. [40]. Bound proteins were eluted from the resuspended cell wall fraction with 500 mM NaCl for 1 h at 4 °C, using overhead shaker, followed by centrifugation at 10,000 *g* at 4 °C. Soluble and salt-eluted proteins (i.e., from cell wall-bound fraction) were washed and concentrated by Centrifugal Filter (Millpore, Darmstadt, Germany) with 50 mM NaOAc buffer pH 5. Protein concentrations were determined by Bradford assay (Roti®-Quant; Roth, Karlsruhe, Germany). Different aliquots were incubated with 6% (*w*/*v*) inulin (Sigma-Aldrich), 1 mM levan (Sigma-Aldrich, St. Louis, MO, USA) or 1–100 mM sucrose (Applichem, Darmstadt, Germany) in 50 mm NaOAc buffer, pH 5.0 at 37 °C for different time intervals. After incubation, the reaction was stopped by heating at 95 °C for 5 min. Released fructose was determined by HPAEC-PAD as described in Wei et al. [41]. In parallel, glucose and fructose were also determined by a coupled spectrophotometric enzyme assay as described in Link et al. [40]. All enzyme measurements were performed under conditions where activities were proportional to enzyme amount and incubation time.

### 4.9. Functional Characterization of Pichia-Expressed FEH and E. coli-Expressed Invertase Inhibitor Proteins

Medium-secreted recombinant FEH protein obtained from transformed *P. pastoris* cultures (see above) was column-purified, dialyzed against 50 mM NaOAc buffer (pH 5), and incubated for 60 minutes at 30 °C with the following substrates: 1–100 mM sucrose for invertase activity, 6% inulin (Sigma-Aldrich, St. Louis, MO, USA) and 1 mM of different fructooligosaccharides (Wako chemicals, Osaka, Japan) for 1-FEH activity, and 1 mM levan (Sigma-Aldrich, St. Louis, MO, USA) for 6-FEH activity. The reaction product fructose was quantified by HPAEC-PAD analysis. Recombinant inhibitor protein Zm-INVINH1 was purified from *E. coli* extracts (see above) by His-tag affinity chromatography [40]. Since obtaining sufficient amounts of correctly folded recombinant inhibitor protein proved to be difficult, inhibitors Zm-INVINH1 were also transiently expressed in leaves of *N. benthamiana* leaves according to Kusch et al. [13]. For extraction of transiently Zm-INVINH1 expressed proteins from *Nicotiana benthamiana* leaves, protein-containing partially purified preparations were prepared via ConA-chromatography and size fractionation followed the protocols [13,36], note that in contrast to acid invertases, inhibitor proteins are not glycosylated. For inhibition assays, recombinant FEH protein was pre-incubated with inhibitor-containing fractions for 30 min at 30 °C to allow complex formation. Thereafter, remaining FEH activity was determined as described above. All enzyme measurements were performed under conditions where activities were proportional to enzyme amount and incubation time.

### 4.10. Carbohydrate Extraction and Analysis

Total soluble carbohydrates were extracted as described by Wei et al. [42,43]. Quantification of glucose, fructose, sucrose, raffinose, stachyose, 1-kestotriose, 1,1-kestotetraose, 1,1,1-kestopentaose, and recording of profiles for inulin and levan, were performed by HPAEC-PAD as described by Wei et al. [41]. For peak identification, glucose (Merck, Darmstadt, Germany), fructose (Sigma-Aldrich), sucrose (Applichem), 1-kestotriose, 1,1-kestotetraose, 1,1,1-kestopentaose (all Wako Chemicals), raffinose (Sigma-Aldrich), and starchyose (Sigma-Aldrich) were used as standards.

### 4.11. Bimolecular Fluorescence Complementation (BiFC) Assay

*Agrobacterium tumefaciens* strain (C58C1) containing the BiFC constructs (see above) were co-transformed into *N. benthamiana* leaves. Yellow fluorescent protein (YFP) signals were detected by confocal laser scanning microscopy. The *Arabidopsis* At1g33811-encoded cell wall protein (C-terminally tagged with cYFP) was used as control.

### 4.12. CLSM Analysis

Microscopic analyses were carried out using a confocal laser scanning microscope (LSM510 Meta, Zeiss, Jena, Germany). The following excitation and detection wavelengths were used: GFP, excitation at 488 nm, detection at bandpass 505–530 nm; RFP, excitation at 543 nm, detection at bandpass 560–615 nm; YFP, excitation at 514 nm, detection at bandpass 530–560 nm. Chlorophyll autofluorescence: Excitation at 488 nm, detection at longpass 650 nm.

### 4.13. Modelling FEH-Invertase Inhibitor Complexes

Homology modeling of Zm-6&1-FEH1 in complex with Zm-INVINH1 was based on the structure of the complex between AtCWI1 and the tobacco invertase inhibitor Nt-CIF [44] (PDB entry 2QXR, https://www.rcsb.org/). All structural alignments and model calculations were performed using the software modeler 9.13 [45]. Protein models were visually inspected with the help of COOT-software version 0.7.1 [46], and high-resolution rendering was performed using Chimera version 1.10.1 (http://www.cgl.ucsf.edu/chimera/) [47].

### 4.14. Statistical Analysis

All gene expression studies via qPCR and all determinations of FEH activity (plant extracts or recombinant proteins) were performed in 3 independent experiments, with 3 technical replicas for each experiment. SD indicates standard deviation of the mean of three independent experiments. For statistical analysis, one-way analysis of variance (ANOVA) followed by a Tukey’s multiple comparison test using SPSS 20.0 software (SPSS Inc., Chicago, IL, USA) were used. For further details see figure legends.

## Figures and Tables

**Figure 1 ijms-20-03807-f001:**
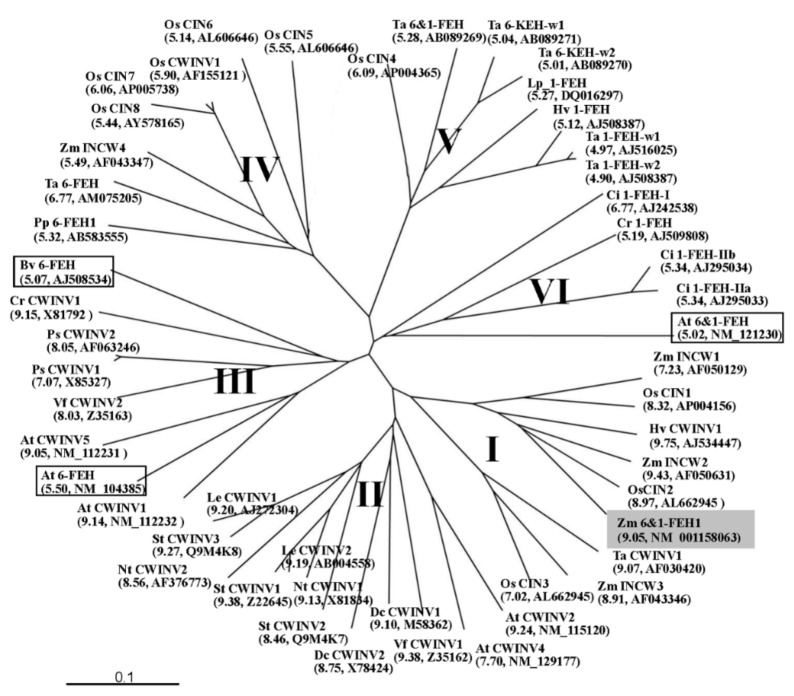
Unrooted phylogenetic tree containing cell wall invertase-like cDNA-derived amino acid sequences and FEHs amino acid sequences. Six groups can be distinguished. Groups (I), (II), and (III) are classified as cell wall invertase (CWIs). CWIs from monocot species are classified in group (I), from dicot species are classified in group (II) and (III). FEHs are classified in group (IV), (V), and (VI). FEHs from dicot species are classified in group (VI), those from monocot species are classified in group (IV) and (V). Functions of rice CWIs (OsCIN4-8) were not confirmed. Zm-6&1-FEH1 is shaded. Three FEHs in non-fructan plants are boxed. Abbreviations for the species are: Ac, *Actinidia deliciosa*; At, *Arabidopsis thaliana*; Bv, *Beta vulgaris*; Cc, *Coffea canephora*; Cr, *Chenopodium rubrum*; Ci, *Cichorium intybus*; Cr, *Campanula rapunculoides*; Dc, *Daucus carota*; Nt, *Nicotiana tabacum*; Hv, *Hordeum vulgare*; Le, *Lycopersicum esculentum*; Lp, *Lolium perenne*; Os, *Oryza sativa*; St, *Solanum tuberosum*; Pp, *Phleum pratense*; Ps, *Pisum sativum*; Ta, *Triticum aestivum*; Vf, *Vicia faba*; Zm, *Zea mays*. Isoelectric points and accession numbers are presented in brackets. pI calculation used software in http://web.expasy.org/compute_pi/. The scale bar indicates branch length. Phylogenetic and molecular evolutionary analysis was conducted with TreeView and MAFFT.

**Figure 2 ijms-20-03807-f002:**
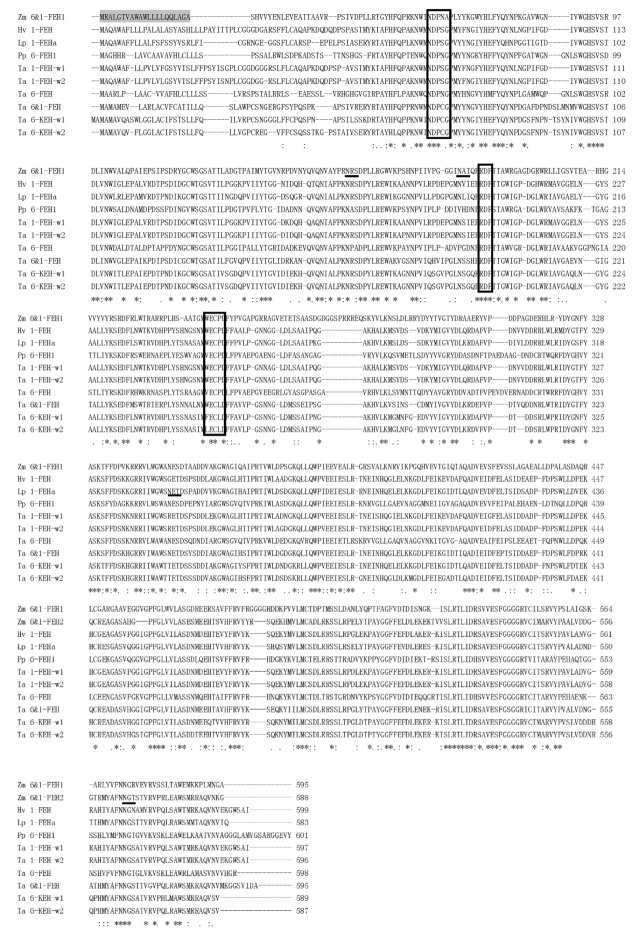
Alignment of amino acid sequence of Zm-6&1-FEH1 with fructan exohydrolases (FEHs) from other *Poaceae* species. β-fructosidase motifs (NDPNG/A), cysteine-containing catalytic sites (MWECP/V), and conserved Asp residues (D) are boxed. Putative glycosylation sites are underlined. Shaded regions represent the predicted N-terminal signal peptides. Asterisks indicate identical residues, colons indicate conserved substitutions, and periods indicate semi-conserved substitutions. Abbreviations for the species are: Hv, *Hordeum vulgare*; Lp, *Lolium perenne*; Pp, *Phleum pratense*; Ta, *Triticum aestivum*; Zm, *Zea mays*. The alignment was generated using CLUSTAL X2 program. Accession numbers of genes used for alignment were: AB089269 (*Triticum aestivum* 6&1-FEH), AM075205 (*Triticum aestivum* 6-FEH), AJ516025 (*Triticum aestivum* 1-FEH w1), AJ508387 (*Triticum aestivum* 1-FEH w2), AB089271 (*Triticum aestivum* 6-KEH w1), AB089270 (*Triticum aestivum* 6-KEH w2), AJ605333 (*Hordeum vulgare* 1-FEH), DQ016297 (*Lolium perenne 1-FEHa*), and BAJ76715 (*Phleum pratense* L. 6-FEH).

**Figure 3 ijms-20-03807-f003:**
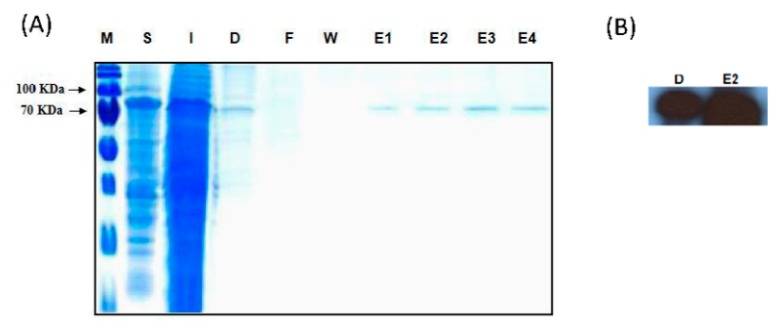
Heterologous expression of C-myc-tagged Zm-6&1-FEH1 protein in *P. pastoris*. (**A**) SDS-PAGE analysis of recombinant Zm-6&1-FEH1 protein. Marker (M), soluble protein (S), insoluble protein (I), dialyzed soluble protein (D), column flow-through (F), wash fraction (W) and elution fractions (E1–E4). (**B**) Immunoblot analysis of dialyzed fraction and eluted Zm-6&1-FEH1 protein, the immunosignal at about 80 kDa being detected with the C-myc antiserum.

**Figure 4 ijms-20-03807-f004:**
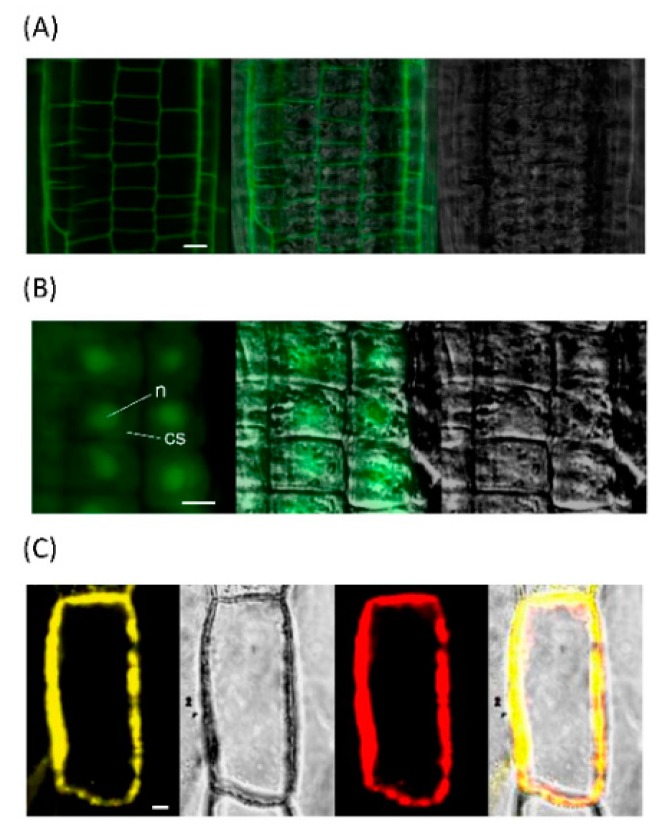
Zm-6&1-FEH1 is targeted to the cell wall. (**A**) The left map represents stable expression of Zm-6&1-FEH1:GFP fusion protein in root cells of *Arabidopsis*, the right and middle maps show bright-field image and overlay. (**B**) The left map represents stable expression of GFP alone in root cells of *Arabidopsis*, the right and middle maps show bright-field image and overlay, n represents nucleus; cs represents cytoplasm. (**C**) Transient expression in onion epidermal cells indicating co-localization of Zm-6&1-FEH1:YFP (left map) and Zm-INVINH1:RFP (third map), the second and last maps show bright-field image and overlay. All images are representative of three independent experiments.

**Figure 5 ijms-20-03807-f005:**
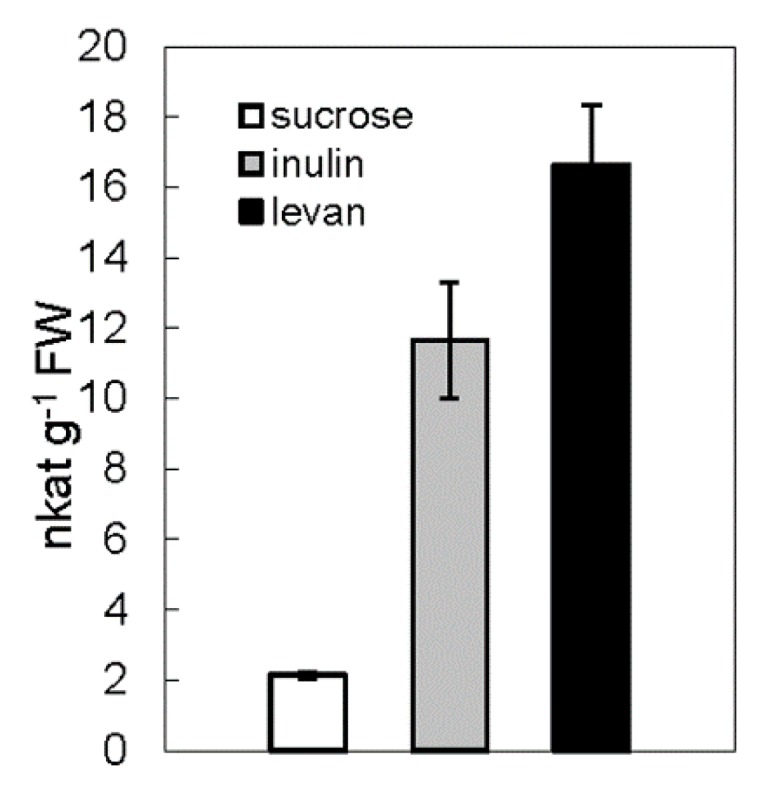
Cell wall-associated invertase and fructan exohydrolase activities in *N. benthamiana* leaves transiently transformed with *Zm-6*&*1-FEH1*. Forty-eight hours after *Agrobacterium tumefaciens*-mediated transformation of *N. benthamiana* leaves, enzyme activities were determined in salt-eluted (1M NaCl) cell wall protein fractions, collected on a Millipore filter with a cut-off of 50 kD. Substrate concentrations were 100 mM sucrose, 1 mM levan, and 6% (*w*/*w*) inulin, respectively. Cell wall invertase activity induced by transformation with empty vector alone was subtracted.

**Figure 6 ijms-20-03807-f006:**
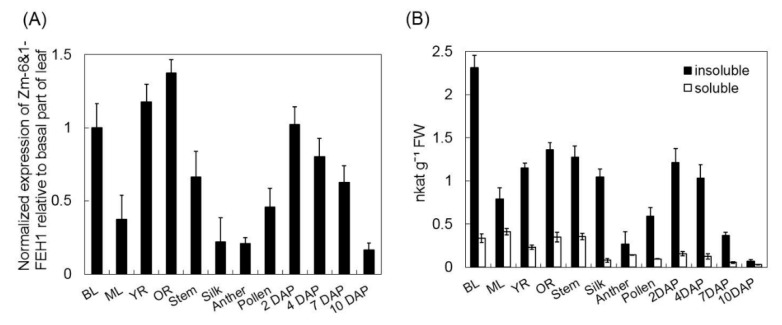
Expression at transcript level for maize 6&1-FEH1 enzyme in different maize tissues as compared to fructan exohydrolase activity. (**A**) Transcript levels for *Zm-6*&*1-FEH1*. Transcript amounts were determined by qPCR. Expression data were normalized with respect to actin and ubiquitin, and are presented relative to basal part of leaf (BL). (**B**) Total FEH activities as determined by enzymatic fructose release in soluble protein fractions and insoluble fractions (i.e., cell wall suspension), respectively, determined with 1 mM levan as substrate. BL, basal part of leaf; ML, middle part of leaf; YR, young root; OR, old root; DAP, days after pollination (seeds). BL, ML, and YR refer to 3-week-old greenhouse cultivated seedlings (at the 4-leaf stage), BL and ML referring to the fourth leaf. All other samples are from fully grown plants at/after pollination. Results are means of 3 biological replicates (± SE), each with 4 technical replicates.

**Figure 7 ijms-20-03807-f007:**
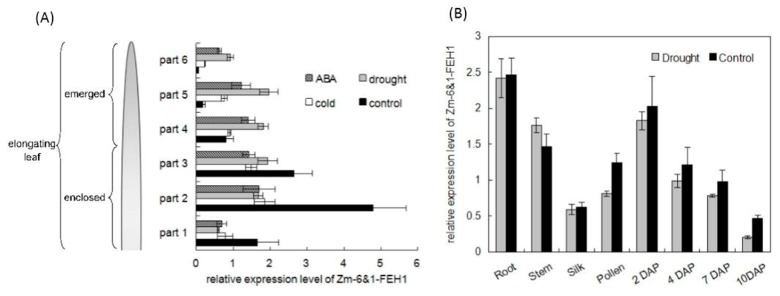
Effect of drought, cold, and abscisic acid (ABA) treatment on the expression of maize 6&1-FEH1 enzyme. Transcript levels for *Zm-6*&*1-FEH1* was determined by qPCR. Expression data are presented relative to part 4 from leaves of control plants to seeds 10 DAP. (**A**) Expression profiles along the axis of the fourth leaf of 3-week-old greenhouse cultivated maize seedlings have been determined under different growth conditions. After identical pre-cultivation for 12 days, subsequent treatments were as follows: Control, regular watering at 3-day-intervals; drought exposure, no additional watering during 9 days; cold exposure, like control but seedlings kept at 4 °C; abscisic acid (ABA) treatment, regular watering as in controls but supplemented with 50 µM ABA. Length of enclosed leaf parts (1–3): 3 cm, length of emerged leaf parts (4–6) 6 cm. (**B**) Expression data from fully grown plants at/after pollination. Control, regular watering at 3-day-intervals; drought, watering was stopped 7 days before pollination.

**Figure 8 ijms-20-03807-f008:**
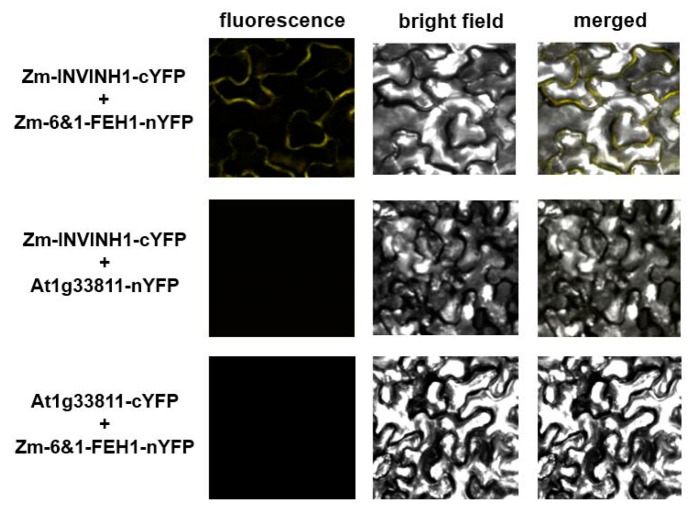
Bimolecular fluorescence complementation (BiFC) analysis reveals complex formation between Zm-6&1-FEH1 enzyme and invertase inhibitor Zm-INVINH1 after transient co-expression in *N. benthamiana* leaves. Invertase inhibitor Zm-INVINH1 was C-terminally fused with cYFP, and Zm-6&1-FEH1 C-terminally fused with nYFP. Inhibitor-cYFP fusion and FEH-nYFP fusion were co-expressed in *Nicotiana benthamiana* leaves for 48 h. Thereafter, leaf epidermal cells were analyzed for fluorescence resulting from YFP complementation. A cell wall-localized protein (encoded by of At1g33811) served as control for cYFP or nYFP fusion.

**Figure 9 ijms-20-03807-f009:**
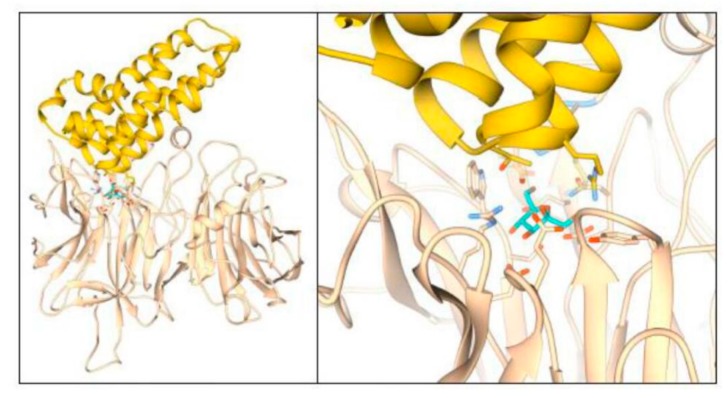
Model for Zm-6&1-FEH1 interaction with Zm-INVINH1. Hypothetical complex modeled on the basis of the crystal structure of AtCWI1-NtCIF (left). The interface is formed by conserved sequence elements of the interacting proteins (right, close up). A second contact site seems to be dispensable for complex stabilization due to absence of residues undergoing stabilizing interactions.

**Table 1 ijms-20-03807-t001:** Specific activities of recombinant maize Zm-6&1-FEH1 expressed in *P. pastoris*. Recombinant enzyme was collected from the culture medium by ammonium sulfate precipitation (80%), followed by dialysis against substrate-free assay buffer. Enzyme activity was determined from released fructose as quantified by HPAEC-PAD analysis. DP, degree of polymerization.

Substrate	Linkage Form	DP	*Pichia* Derived Activity of Zm-6&1-FEH1 (%)
6% inulin	β (2,1)	30*	85
1 mM 1-kestotriose	β (2,1)	3	45
1 mM 1,1-kestotetraose	β (2,1)	4	39
1 mm 1,1,1-kestopentaose	β (2,1)	5	36
1 mM levan	β (2,6)	100*	100
1 mM sucrose		2	21
10 mM sucrose		2	20
100 mM sucrose		2	20

* Mean DP.

**Table 2 ijms-20-03807-t002:** Effect of maize invertase inhibitor transiently expressed in leaves of *N. benthamiana* on enzyme activity of recombinant maize FEH expressed in *P. pastoris*, as compared with control extracts from *N. benthamiana* leaves. Before addition of substrate, enzyme and inhibitor-containing extract were pre-incubated for 30 min. Enzyme activity is given as percentage of the control value. Significant inhibition of enzyme activity is shaded. Values are means ± SD (*n* = 3).

Zm-6&1-FEH1	**Control**		**Zm-INVINH1**	
	inulin	levan	inulin	levan
	100 (±10)	100 (±8)	66 (±11)	64 (±8)

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
