# Peer review of "A 6&1-FEH Encodes an Enzyme for Fructan Degradation and Interact with Invertase Inhibitor Protein in Maize (Zea mays L.)"

_ijms, 2019, doi:10.3390/ijms20153807_

Round 1
Reviewer 1 Report
Introduction
Some acronyms used are not defined. Eg: “VI” line 42.
Some typographical errors are noticed within the manuscript – these are minor and would assume be fixed upon editing. Eg: Line 129, “idnetify”
Results
The term “DP” in table 1 is not defined, please define this in table description.
Figure 4: Images in 4c are not defined. Which one is which in the four images?
The authors say that Zm-6&1-FEH1 is localised to the cell wall (line 135). Figure 4 says Apoplast. Which is it? Perhaps a positive control is needed to be shown here.
Line 199: “Specific functionalities”. Too vague – remove this.
Lines 225-232: Remove methodology from this section, it is not necessary to have this here.
Methods:
Line 340: Surely infiltrated leaves where not used immediately. Please specify how long you left the tissues post-infiltration to allow expression of the vectors.
Line 368-370: The n and c terminals of split YFP fused to the n or c terminal of the protein of interest? Did the cDNA consist purely of coded regions or also UTRs??
Lines 456-458: There is no details on statistical analysis here, only replicates. Add in details on stats analysis here (SPSS, excels, students, anova etc,).
Discussion:
I do have a slight problem with the experiments looking at inhibition of the Zm-6&1-FEH1 by Zm-INVINH1. (Table 2). There should be an inhibition curve for this, this is standard practice for enzyme inhibition studies. However given the other experiments and the nature of discussion the data presented is fine.
Although the results do not directly show an interaction between Zm-6&1-FEH1 and Zm-INVINH1, the authors provide sufficient evidence to suggest this interaction.
The authors discuss their work appropriately and fit it within the current literature. Their results are of course speculative as to the function or strength of inhibition but they provide evidence for the mechanisms they suggest.
Reviewer 2 Report
This is an excellent manuscript that describes a study conducted on fructan degradation mediated by the enzyme fructan exohydrolase. The study is planned well, experimental approaches are thorough and state of the art. I would consider the manuscript as a novel contribution,in the field of carbogydrate research!
Author Response
Thank you very much for your comments!